# PARALLAX: EFFICIENT LLM INFERENCE SERVICE OVER DECENTRALIZED ENVIRONMENT

## ABSTRACT

Deploying a large language model (LLM) inference service remains costly because centralized serving depends on specialized GPU clusters and high-bandwidth interconnects in datacenters. An appealing alternative is to leverage collaborative decentralized GPU pools. However, heterogeneity in GPU and limited interconnected network bandwidth, along with potentially dynamic availability, make efficient scheduling the central challenge in this scenario. In this paper, we present PARALLAX, a decentralized LLM serving system that turns a pool of heterogeneous GPUs into an efficient inference platform via a two-phase scheduler. PARALLAX decomposes planning into (**i**) model allocation, which places layers of each replica across diverse GPUs to jointly optimize latency and throughput under memory and link-bandwidth constraints, and (**ii**) request-time GPU pipeline selection, which stitches layers from different replicas into end-to-end execution chains that balance load and adapt to current conditions. We implement PARALLAX and evaluate it on open-source LLMs deployed over real volunteer nodes. PARALLAX consistently reduces latency and increases throughput relative to decentralized baselines, demonstrating that principled scheduling can make volunteer compute a practical, affordable substrate for LLM inference.

## 1 INTRODUCTION

Large language models (LLMs) have demonstrated unprecedented capabilities across a wide range of applications, yet deploying the generative inference service as an AI application infrastructure remains prohibitively resource-intensive. The current centralized deployments demand specialized GPU clusters, high-bandwidth interconnects, along with substantial operational budgets, which place LLM services beyond the reach of many organizations. A recent promising alternative is to harness decentralized collaborative computing resources — specifically, a pool of heterogeneous volunteer GPU nodes that contribute idle GPU compute — to support LLM inference at substantially reduced cost (Yuan et al., 2022; Borzunov et al., 2023). This paradigm holds the potential to democratize access to advanced AI services by transforming unused, globally distributed compute into a shared infrastructure. In this paper, we explore *how to enable efficient LLM inference service in a real-world decentralized collaborative environment through effective scheduling*.

Despite its promise, decentralized collaborative inference introduces a fundamentally different set of design considerations compared to conventional datacenter deployments. The core compelling issue lies in the tension between the democratizing promise of decentralized inference and the practical difficulties of realizing it at scale. Volunteer GPU nodes are inherently diverse in their compute power, memory capacity, network connectivity, and reliability. Unlike homogeneous datacenter clusters, such environments introduce variability that can easily undermine inference service quality and user experience. On the other hand, the potential scale of globally available idle GPUs represents a massive untapped resource: even partial utilization could significantly lower the barrier to deploying advanced AI services. Bridging this gap requires effective strategies for scheduling that can transform a highly heterogeneous, decentralized infrastructure into an economically reliable platform for LLM inference (Borzunov et al., 2023; Wu et al., 2025).

Deploying LLM inference directly over a decentralized pool of volunteer GPUs is nontrivial because straightforward extensions of datacenter solutions encounter some essential challenges. First, most existing inference frameworks (Kwon et al., 2023; Zheng et al., 2024) assume homogeneous

hardware and rely on symmetric parallelization schemes, but in decentralized settings, GPU nodes differ drastically in FLOPs, memory, and bandwidth. Naively partitioning a model evenly across such devices causes stragglers, where slower GPUs throttle the entire pipeline while faster ones remain underutilized. Second, communication heterogeneity further complicates the scheduling: links between volunteer nodes can span from high-speed local connections to congested, cross-region networks. Standard collective operations, which are efficient in datacenter clusters, degrade significantly under these conditions. Thus, directly applying datacenter-oriented strategies is far from enough — new scheduling and coordination mechanisms are required to fully unlock the potential of decentralized collaborative LLM inference.

Due to these challenges, prior attempts have only partially explored decentralized LLM inference serving. The Petals (Borzunov et al., 2023) system is a pioneering effort that allows users to collaboratively host large models by contributing their GPUs in a swarm, where it demonstrated the feasibility of volunteer-driven LLM inference, but it relies on swarm-parallel (Ryabinin et al., 2023) coordination with greedy scheduling heuristics and no global optimization — Consequently, Petals cannot fully account for device heterogeneity or network bottlenecks, often leading to under-utilization of available resources. In practice, Petal's performance would be frequently limited by the slowest participants and suboptimal LLM layer placements.

In this paper, we introduce PARALLAX, an efficient LLM inference system designed for decentralized collaborative environments. At the core of PARALLAX is a novel scheduling algorithm that allocates layers of the LLM serving replicas across participating GPU nodes in a way that adapts to each GPU's compute power and their interconnected network bandwidth. Towards this end, we propose a two-phase scheduler that produces a decentralized serving plan for LLM inference over volunteer, peer-to-peer GPUs connected by heterogeneous, low-bandwidth links. The plan has two components: (**i**) a model allocation strategy (as Phase-1 scheduling) that places layers of each LLM model replica across diverse GPUs to minimize per-request latency while maximizing system throughput; and (**ii**) a GPU pipeline-chain selection policy (as Phase-2 scheduling) that, at request time, stitches layers from different replicas of pipeline stages into an end-to-end pipeline to balance load and improve utilization. Because jointly optimizing placement and routing would be NP-hard under heterogeneity, Phase 1 scheduling uses dynamic programming with a water-filling heuristic to compute latency- and throughput-aware allocations, and Phase 2 scheduling treats the resulting placement as a directed acyclic graph (DAG) and applies dynamic programming to select per-request chains that adapt to current resource conditions. Together, these phases deliver low latency and high throughput for a decentralized LLM inference service.

To evaluate the effectiveness of the design and implementation in PARALLAX, we present an extensive experimental evaluation of Parallax on open-source LLMs (including models up to tens of billions of parameters) deployed over real distributed volunteer nodes. The results demonstrate that PARALLAX significantly outperforms existing decentralized heterogeneous inference baselines in terms of both throughput and latency. Concretely, we achieve up to $3.6\times$ (average $1.58\times$) higher throughput and up to $3.2\times$ (average $1.66\times$) lower latency across different models, workload traces, and request arrival rates. These findings validate that an open collaborative pool can serve LLMs efficiently when guided by PARALLAX 's scheduling and system optimizations, marking an important step toward affordable, democratized LLM deployment.

## 2 PRELIMINARIES AND RELATED WORK

**LLM generative inference.** Inference for autoregressive LLMs comprises two phases: *prefill* and *decoding*. During prefill, the prompt is processed in a single forward pass to construct the key–value (KV) cache and to emit the first output token; this phase exhibits high arithmetic intensity and is therefore compute-bound. Decoding then proceeds iteratively, generating a new token based on the KV cache per step; because each step performs a small compute load while reading and writing a large volume of KV cache, which makes this phase predominantly limited by high-bandwidth memory (HBM) I/O. Some standard strategies are applied to parallelize the LLM inference computation: Request-level parallelism assigns distinct requests to independent model replicas (which can be viewed as a form of data parallelism), incurring no inter-replica communication on the inference critical path. To scale a single model across devices, pipeline parallelism (PP) (Huang et al., 2019; Ryabinin et al., 2023) partitions the network into stages mapped to separate GPUs (or GPU groups)

and communicates only inter-stage activations. Compared with tensor model parallelism Narayanan et al. (2021), mixture-of-experts parallelism Fedus et al. (2022), and sequence parallelism Liu et al.; Wu et al. (2024), PP typically incurs much lower communication volume because parameters and per-token KV shards remain local to each stage while only activations traverse the pipeline.

**Decentralized ML platform**. Recent works have explored how to harness decentralized and heterogeneous compute resources for machine learning computation. Early volunteer-computing paradigms (e.g., SETI@home Anderson et al. (2002)) demonstrated the power of aggregating idle personal devices for large computational tasks. Following this spirit, researchers have recently proposed distributing ML workloads across collaborative networks of heterogeneous GPUs, edge devices, or multiple clouds Jiang et al. (2024; 2025b); Mei et al. (2025); Yuan et al. (2022); Wang et al. (2022). For example, some works envision training foundation models on sparsely used edge accelerators or idle edge GPUs to improve sustainability and resource utilization Yuan et al. (2022); Wang et al. (2022); Miao et al. (2023); Xue et al. (2025). Furthermore, Sky computing Stoica & Shenker (2021); Yang et al. (2023) vision promotes using multiple cloud providers as one federated platform. These efforts, along with frameworks for decentralized AI services like SAKSHI Bhat et al. (2023), which uses a blockchain proof-of-inference layer to incentivize volunteer GPUs and ensure trust, highlight a growing interest in distributed ML. In the context of generative model inference, only a few systems tackle truly decentralized execution Borzunov et al. (2023); Fang et al. (2025). One interesting attempt is Petals Borzunov et al. (2023), which enables collaborative inference and fine-tuning of large language models by pooling volunteer GPUs. Petals partitions a model (e.g., BLOOM-176B Workshop et al. (2022)) across participants and coordinates inference as a pipeline of peer-to-peer computations. Petals' design relies on swarm parallelism Ryabinin et al. (2023), where peers communicate in a distributed network (DHT) to sequentially execute model layers. However, Petals lacks a global scheduling mechanism for inference requests, which could lead to compromised performance for a high-quality LLM inference service.

# 3 DECENTRALIZED SCHEDULING OF LLM INFERENCE SERVICE

This section introduces our scheduling algorithm for decentralized LLM inference serving.

## 3.1 SCHEDULING OVERVIEW

To accommodate the LLM inference service in decentralized and heterogeneous scenarios, our scheduling algorithm must determine two essential strategic components:

- **Model allocation strategy**: The model allocation strategy determines how to allocate the model layers in each model replication across a heterogeneous set of GPUs connected by heterogeneous low-bandwidth networks. An optimal allocation would minimize inference latency for each request and maximize overall system serving throughput.
- **GPU pipeline chain selection strategy**: Given the pre-allocated model replicas, the GPU pipeline chain selection strategy determines the actual executed GPU pipeline chain that dynamically constructs a complete model from different model replicas to serve each inference request. An effective pipeline chain selection strategy balances the workload among pipelines, further enhancing system and resource utilization.

These two components together form a *decentralized serving plan*. Note that, given the potentially large search space, determining the optimal model allocation and GPU chain selection strategies would be NP-hard. The *heterogeneous* and *dynamic* nature of the decentralized scenario, therefore, demands an *effective* and *efficient* scheduling approach. To meet this need, we design a two-phase scheduling algorithm: (**i**) The first phase (§3.2) employs a dynamic programming and water-filling algorithm informed by empirical heuristics to search for an optimal model allocation, and (**ii**) the second phase (§3.3) treats that allocation as a DAG and uses dynamic programming to select the optimal GPU chains for individual clients.

## 3.2 PHASE 1 SCHEDULING: MODEL ALLOCATION

We first determines how to allocate the multiple replicas of an LLM model with $L$ layers, indexed by $\{1, 2, \ldots, L\}$, across a set of $N$ heterogeneous GPU noted by $\mathbf{G} = \{g_1, g_2, \ldots, g_N\}$, where $g_i$

represents the $i$-th GPU. In a decentralized cluster, request data parallelism (i.e., layer replication) and pipeline model parallelism are used to distribute and parallelize model layer execution across GPUs; each GPU is responsible for serving one pipeline stage (i.e., a continuous set of model layers). One example of the first phase model allocation is shown in Figure 1. To guide this search efficiently, we introduce a heuristic that exploits fundamental characteristics of decentralized inference.

**Key observations and derived heuristics.** In decentralized scenarios, inter-region GPU connections exhibit ultra-low bandwidth (down to hundreds of MB per second), creating significant communication overhead for decentralized LLM inference serving. Concretely, the communication time between consecutive pipeline stages could be significantly longer than the GPU computation time (i.e., the pipeline stage execution time). Based on this key observation, we design two heuristics:

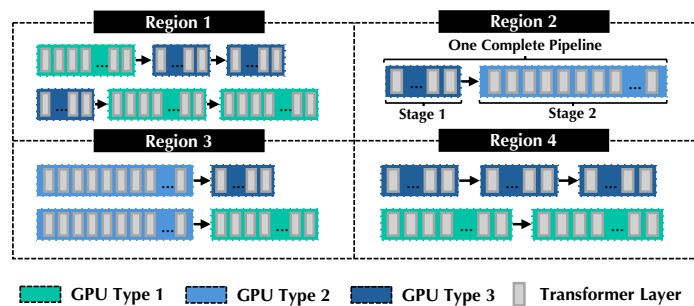

Figure 1: Example of the first phase model allocation among heterogeneous GPU types across different geographic regions.

- *Region-based heuristic*: We force Phase-1 scheduling to operate on a per-region basis, constraining layer allocation within regional boundaries to minimize cross-region data transfer and optimize serving efficiency.

- *Latency-dominant heuristic*: We force Phase-1 scheduling to prioritize the solution with a smaller number of pipeline stages. Note that each additional pipeline stage in a pipeline introduces communication overhead that significantly outweighs the benefits of distributing the model computation across more devices, while each additional pipeline replication proportionally increases the number of requests that can be served concurrently.

Comprehensively, this observation and the derived heuristics materialize our first phase scheduling objectives: (**i**) Minimize the number of stages in each pipeline to reduce execution latency, and (**ii**) maximize the number of pipeline replications to increase system throughput.

**Heuristic-based dynamic programming algorithm.** To perform Phase-1 scheduling, we propose a *dynamic programming algorithm* that implements the region-based and latency-dominant heuristic to obtain a model allocation strategy that minimizes pipeline inference latencies while maximizing overall system throughput. We define $c_i \in \mathbb{N}$ [1] to be the maximum layer capacity of GPU $g_i$, $k$ to be the number of pipeline replications, and $s^\star(k)$ to be the minimum total number of stages required to accommodate $k$ pipeline replications. Our objective is to maximize the number of replications $k$ while minimizing the average stages per replication $s^\star(k)/k$. The procedure follows three steps:

(**i**) P1-*Initialization*: The algorithm sorts GPU layer capacities in non-increasing order to obtain $\mathbf{c} = (c_1 \geq \cdots \geq c_N)$, and computes the maximum possible replication number $k_{\max} = \min(N, \lfloor(\sum_{i=1}^N c_i)/L\rfloor)$. It initializes a dynamic programming state for Phase 1 scheduling noted by $\mathrm{dp}_1(0, \emptyset, 0)$ for each $k \in \{1, \ldots, k_{\max}\}$ with an empty multiset of residuals for partially assigned pipelines, zero fully assigned pipelines, and a companion table of back-pointers.

(**ii**) P1-*DP exploration*: The dynamic programming state $\mathrm{dp}_1(i, \mathbf{r}, f)$ represents the assignment status when processing GPU $g_i$ (with capacity $c_i$) for target replication count $k$. The state tracks $\mathbf{r} = (r_1 \leq r_2 \leq \cdots \leq r_m)$ as the sorted residual layer counts for partially assigned pipelines, where each $r_j \in \{1, 2, \ldots, L-1\}$, and $f$ as the count of fully assigned pipelines (containing all $L$ layers). At each GPU indexed by $i$, the algorithm considers three transitions:

- ❶ Skip GPU: Transition to $\mathrm{dp}_1(i+1, \mathbf{r}, f)$ without assigning the $i$-th GPU to any pipeline.

---

[1]Consistent with prior work (Borzunov et al., 2023), we define the layer capacity of each GPU as the maximum number of transformer layers that can be loaded into its available VRAM, while ensuring a small memory budget is reserved for activations and the KV cache.

- ❷ Extend existing pipeline: Select a partially assigned pipeline $j$ and assign the $i$-th GPU to this pipeline. Update the residual count as $r_j \leftarrow r_j - c_i$. If $r_j \leq 0$, the pipeline becomes fully assigned (increment $f$ and remove $r_j$ from $\mathbf{r}$).

- ❸ Start new pipeline: Create a new pipeline starting with the $i$-th GPU, subject to the constraint $f + |\mathbf{r}| < k$. Initialize residual count $r = L - c_i$. If $r \leq 0$, the pipeline is immediately fully assigned (increment $f$); otherwise, add $r$ to $\mathbf{r}$.

The algorithm evaluates all valid transitions, records the one yielding the minimum number of pipeline stages, and stores the corresponding decision pointer for backtracking.

(**iii**) P1-*Objective evaluation and reconstruction*: The algorithm sets $s^\star(k) = \mathrm{dp}_1(0, \emptyset, 0)$ and, for each $k \in \{1, \ldots, k_{\max}\}$, computes $Z(k) = k^\alpha / \big(T_{\mathrm{comp}} + (s^\star(k)/k)\, r_{\mathrm{RTT}}\big)$. Note that $\alpha > 0$ controls how strongly the score favors additional replications relative to the per-replication latency term, $T_{\mathrm{comp}}$ is the average per-replication compute time (excluding communication), and $r_{\mathrm{RTT}}$ is the average inter-stage hop latency obtained from profiling. The algorithm then selects $\hat{k} = \arg\max_k Z(k)$, backtracks decisions to recover GPU-to-pipeline assignments, and emits contiguous layer blocks per stage in pipeline order using a write cursor to ensure gap-free layer placement.

**Water-filling algorithm for intra-pipeline rebalancing.** Heterogeneity in GPU compute power may result in imbalanced pipeline stage execution times after the dynamic programming step. To mitigate this, we employ a *water-filling algorithm* that redistributes layers to balance stage times while preserving both contiguous-layer order and GPU assignment. In a pipeline replication $\mathcal{P} = (g_{i_1}, \ldots, g_{i_n})$, each GPU $g_i$ has effective layer capacity $c_i$ and compute capacity $F_i \in \mathbb{R}^+$. We model each GPU as a container filled proportionally to $F_i$ until the total allocation reaches $L$. The water-filling algorithm consists of three steps: (**i**) Fractional layer allocations $x_i = \min(c_i, \lambda F_i)$ are computed for each GPU using a scaling parameter $\lambda > 0$ determined via binary search. (**ii**) The fractional allocations are rounded to integer allocations $x_i^{\mathrm{int}} \in \mathbb{N}$ using the largest remainder (Hamilton) method (Balinski & Young, 2010), subject to the layer capacity constraints $c_i$. (**iii**) The integer allocations are mapped to consecutive layer intervals in pipeline order, ensuring that all $L$ layers are covered exactly once without gaps or overlaps. Consequently, each GPU is assigned layers aligned with its compute capacity under layer capacity constraints, maximizing utilization and optimizing stage execution times across the pipeline.

The dynamic programming and water-filling rebalancing approach guarantees both efficient search and optimal scheduling results. Note that the Phase 1 scheduling algorithm does not account for load balancing across pipelines, and it allows different pipeline replications to exhibit performance gaps due to GPU heterogeneity; this issue is further discussed in the Phase 2 scheduling (§3.3).

## 3.3 Phase 2 Scheduling: GPU Pipeline Chain Selection

The second phase of scheduling assigns, to each client, a concrete GPU pipeline chain that will run the entire model end-to-end. Starting from the layer placement produced in the first phase, we build a layer-indexed DAG whose nodes $(\ell, g_i)$ denote replications of layer $\ell$ stored on GPU $g_i$. A single left-to-right dynamic programming sweep over this DAG finds the minimum latency path from the first to the last layer; the resulting path is recorded as the client's GPU chain. One example of the second phase GPU pipeline chain selection is shown in Figure 2.

**Distributed hash table.** A distributed hash table (DHT) is required for the second phase to store the live performance map, which captures the processing latency and real-time RTT for each GPU within the cluster. When a GPU joins the cluster, it declares two essential attributes to the DHT: (**i**) Node ID: a DHT hash used for request routing, and (**ii**) RAM capacity: indicating the number of KV tokens each layer can hold, which informs the computational resources allocated to each layer. During runtime, each GPU publishes two key families to the DHT: (**i**) $\tau_{<g_i, \ell>}$, the profiled layer-level latency over GPU $g_i$, and (**ii**) $\rho_{<g_i, g_{i'}>}$, the one-way RTT between GPU pair $g_i$ and $g_{i'}$. The profiling and publishing processes occur periodically (every 1-2 seconds). When selecting a client chain, $\tau_{<g_i, \ell>}$ serves as node weights and $\rho_{<g_i, g_{i'}>}$ as edge weights in the DAG, enabling the dynamic programming sweep to return the true minimum latency path and naturally deflect traffic toward less loaded pipelines, thereby resolving the load balancing issue mentioned in the first phase. Each time a GPU pipeline chain is selected or released, the GPUs on that pipeline chain immediately update their new $\tau_{<g_i, \ell>}$ values, so the DHT always reflects the cluster's current load.

**Pipeline chain selection constraint.** The dynamic programming search is run under one hard constraint: The chosen path must cover *every* model layer exactly once and in order; if layer $\ell$ lies on GPU $g_i$ and layer $\ell+1$ is assigned to GPU $g_{i'}$, then $\ell$ and $\ell+1$ must be consecutive indices in the DAG. This constraint guarantees that every selected GPU chain forms a legal, gap-free pipeline replication, and respects the contiguous-slice model allocation fixed in the first phase.

**Dynamic programming search.** Given the live performance map stored in the DHT, the system employs a *dynamic programming algorithm* to identify the optimal GPU chain that minimizes end-to-end inference latency. Since the DAG constructed from the contiguous-slice allocation is inherently topologically ordered by layer index, a single forward pass

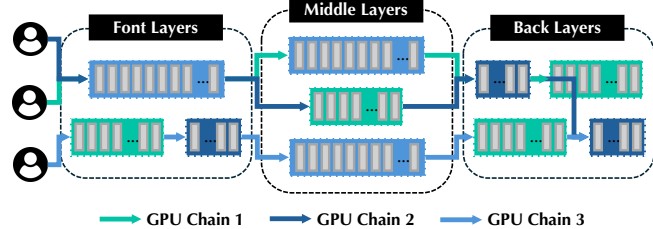

Figure 2: Example of the second phase GPU pipeline chain selection among GPUs (pipeline stages).

suffices to compute the minimum latency path while respecting the chain selection constraint. The algorithm proceeds in three steps:

(**i**) P2-*Initialization*: The cost table is initialized by setting Phase 2 scheduling dynamic programming state $\mathrm{dp}_2(1, g_i) = \tau_{<g_i,1>}$ for every GPU $g_i$ that contains the first model layer, with null back-pointers stored for each entry. This establishes the base case where inference begins at layer 1 with the profiled processing latency $\tau_{<g_i,1>}$ as the initial cost.

(**ii**) P2-*DP propagation*: The algorithm performs a single sweep from layer 1 to layer $L-1$. For each valid edge $(\ell, g_i) \rightarrow (\ell+1, g_{i'})$ in the DAG, the algorithm applies the relaxation step: $\mathrm{dp}_2(\ell+1, g_{i'}) = \min\big(\mathrm{dp}_2(\ell+1, g_{i'}), \mathrm{dp}_2(\ell, g_i) + \rho_{<g_i,g_{i'}>} + \tau_{<g_{i'},(\ell+1)>}\big)$, where $\mathrm{dp}_2(\ell, g_i)$ represents the minimum cumulative latency to reach layer $\ell$ on GPU $g_i$, $\rho_{<g_i,g_{i'}>}$ captures the communication overhead between GPUs, and $\tau_{<g_{i'},\ell+1>}$ accounts for the processing latency of layer $\ell+1$ on the destination GPU $g_{i'}$. When a relaxation updates the minimum cost, the algorithm records the parent GPU $g_i$ that achieved this improvement, enabling path reconstruction.

(**iii**) P2-*Optimal path extraction*: Upon reaching the final layer $L$, the algorithm selects the GPU $\hat{g} = \arg\min_{g_i} \mathrm{dp}_2(L, g_i)$ that yields the minimum total latency. By backtracking through the stored parent pointers, the algorithm reconstructs the complete GPU chain, which is guaranteed to form a gap-free, consecutive pipeline that satisfies the chain selection constraint. This optimal chain is then pinned to the client's session for the duration of the inference request.

**Complexity analysis.** The computational efficiency of the dynamic programming algorithm derives from its single-pass traversal, wherein each edge in the DAG is examined exactly once. This systematic approach yields $\mathcal{O}(L\bar{R}^2)$ time complexity and $\mathcal{O}(L\bar{R})$ space complexity, where $\bar{R}$ denotes the average number of candidate copies per layer admitted into the DAG. This algorithm enables real-time path selection under the current DHT snapshot, ensuring optimal route computation even within large-scale heterogeneous clusters comprising hundreds of GPUs.

### 3.4 HANDLING DYNAMIC MEMBERSHIP

In decentralized collaborative environments, the dynamic entry and exit of GPUs necessitate the design and implementation of robust mechanisms to maintain system coherence without disrupting service. Both phases of our scheduling algorithm are designed to adapt gracefully to these membership changes while preserving optimal performance.

**Phase-1 scheduling adaptation.** The first phase manages membership changes through targeted adjustments that minimize system disruption. The system responds to membership changes through three mechanisms:

(**i**) *GPU joining*: When a new GPU $g_{\mathrm{new}}$ joins, the system consults the DHT's live performance map to identify the bottleneck layer $\ell^*$ with minimum RAM capacity (i.e., minimum computational resources are allocated to this layer), then greedily assigns a contiguous slice of layers $[\ell^*, \ell_{\mathrm{end}}]$ subject to the GPU's layer capacity constraint $c_{\mathrm{new}}$, and immediately republishes the updated RAM capacity to the DHT.

(**ii**) *GPU leaving*: When a GPU leaves, the system de-allocates its assigned slice of layers $[\ell_a, \ell_b]$ and withdraws its performance metrics from the DHT.

(**iii**) *Global rebalancing*: The system triggers global rebalancing (i.e., re-runs the first phase scheduling algorithm and reallocates model layers based on the new scheduling outcomes) when either:

- ❶ The system does not have a full pipeline covering $[0, L)$ (i.e., some layers remain unassigned).
- ❷ The coefficient of variation of layer loads [2] exceeds a configurable threshold, indicating significant imbalance.

Otherwise, the system retains only localized adjustments from operations (**i**) and (**ii**).

**Phase-2 scheduling adaptation.** The second phase adapts to membership changes implicitly through the DHT's temporal consistency mechanisms. A newly joined GPU immediately begins publishing its performance metrics, $\tau_{<g_i,\ell>}$ and $\rho_{<g_i,g_{i'}>}$, making it eligible for selection in subsequent dynamic programming iterations. When a GPU departs, its associated keys in the DHT expire according to the established profiling and publishing intervals (§3.3) and are automatically purged. This prevents the routing algorithm from directing future clients to obsolete GPUs.

Together, these two adaptation mechanisms ensure that (**i**) the model allocation maintains high performance efficiency, and (**ii**) the clients are always routed along the lowest latency paths. When membership changes necessitate reallocation, only the affected GPUs undergo weight reloading or eviction. As a result, any temporary throughput degradation is confined to a small subset of GPUs, and ongoing inference traffic experiences minimal disruption.

## 4 PARALLAX EVALUATION

To evaluate the design and implementation of PARALLAX, we ask the following essential questions:

- *RQ1: How does the end-to-end performance of our proposed system compare to state-of-the-art heterogeneous serving frameworks in terms of latency and system throughput?*
- *RQ2: How effective is our proposed scheduling algorithm in real-world deployment scenarios?*

### 4.1 EXPERIMENTAL SETUP

**Decentralized environment.** We establish a heterogeneous distributed computing environment by provisioning heterogeneous GPU resources across multiple data centers interconnected via public networks. The testbed comprises 5 `RTX 5090` machines and 2 `RTX 4090` machines distributed across geographically separated data centers, with an average inter-machine communication latency of 10 ms over the public network infrastructure.

**Baseline.** We compare PARALLAX with HexGen (Jiang et al., 2024; 2025b), a state-of-the-art decentralized heterogeneous LLM inference engine. HexGen employs static load partitioning over heterogeneous and decentralized environments, leveraging tensor and pipeline parallelism to distribute model execution. HexGen also integrates a scheduling algorithm to optimize resource allocation and reduce inference latency in distributed, heterogeneous, and decentralized environments.

**Models and traces.** We evaluate PARALLAX on Qwen3-32B (Yang et al., 2025) of different precisions: BF16 and FP8 (16B parameters), a representative family of popular open-source transformer models. Following prior work (Mei et al., 2025), we generate workload traces based on real-world data. Our testing traces are subsampled from the ShareGPT and WildGPT datasets, well-known collections of user conversations with ChatGPT that provide realistic inference workload patterns.

**Evaluation metrics.** Following previous evaluation setups (Jiang et al., 2025a), we evaluate system performance based on overall throughput and various percentile latencies (i.e., average, p95, ..., p99, p100 latencies). In particular, the p95 latency denotes the maximum response time within

---

[2]Load is computed as: $\alpha \cdot \frac{\text{current\_kv\_size}}{\text{total\_cluster\_memory}} + (1-\alpha) \cdot \frac{\text{current\_compute}}{\text{total\_cluster\_flops}}$, where $\alpha$ denotes how much we value GPU memory against compute power (set to 0.5 by default), `current_kv_size` is the sum of KV cache memory from all GPUs hosting this layer, and `current_compute` is the sum of FLOPs from all GPUs hosting this layer. Return 0 if total cluster resources are zero.

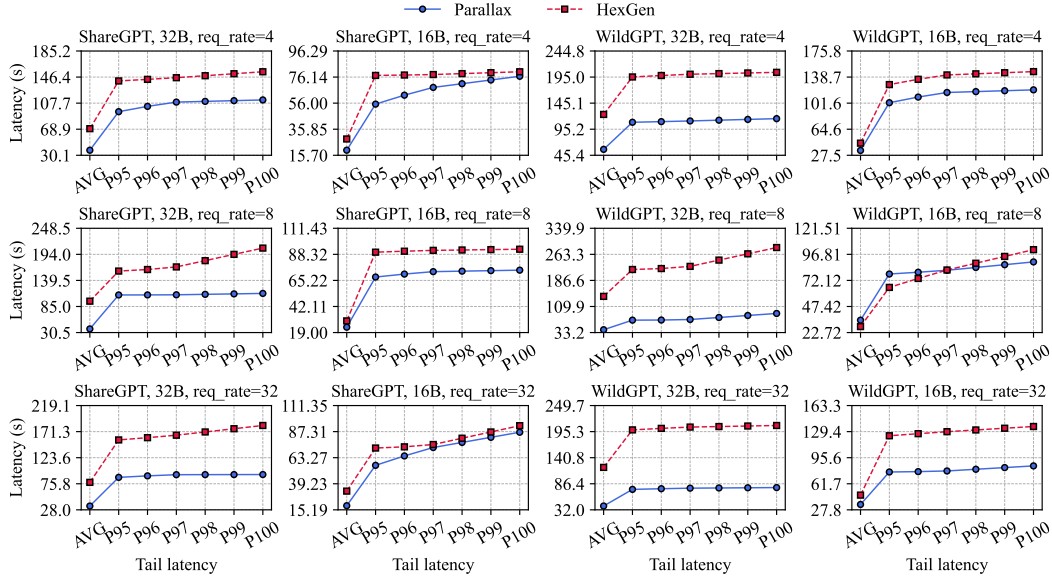

Figure 3: End-to-end latency comparison between PARALLAX and HexGen across different models, traces, and request arrival rates.

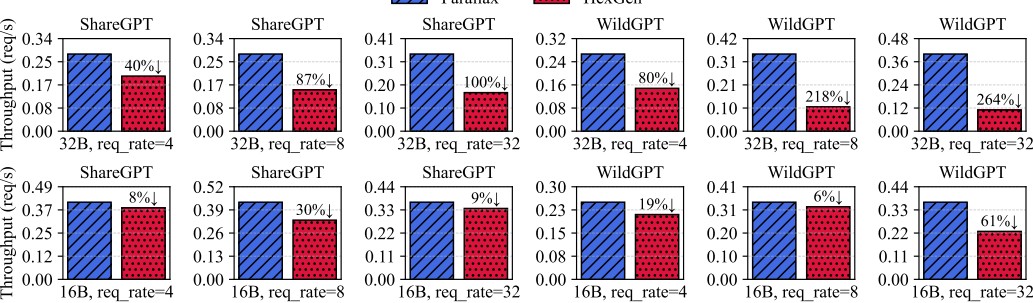

Figure 4: End-to-end throughput comparison between PARALLAX and HexGen across different models, traces, and request arrival rates.

which 95% of all requests are completed. This combination of throughput and tail latency metrics provides a comprehensive view of system performance under different operational conditions.

## 4.2 END-TO-END PERFORMANCE EVALUATION (RQ1)

**Latency comparison between PARALLAX and HexGen.** Across all settings, PARALLAX consistently outperforms HexGen in terms of average and tail latency (p95–p100). For example, under the ShareGPT trace with FP16 models at a request rate of 4, PARALLAX achieves an average latency of 37.6 ms and p99 latency of 111.4 ms, compared to HexGen's 69.7 ms and 151.4 ms, yielding a 1.36× speedup at the tail. The advantage becomes more pronounced in the WildGPT trace, where at a request rate of 32 with FP16, PARALLAX reduces the p99 latency from 207.0 ms (HexGen) to 78.1 ms, corresponding to a 2.6× improvement. Even for lighter FP8 configurations, PARALLAX sustains lower tail latencies: on ShareGPT with a request rate of 32, p99 latency is 82.1 ms under PARALLAX versus 87.0 ms under HexGen, while on WildGPT with a request rate of 8, the gap widens to 87.0 ms versus 94.8 ms. These results indicate that HexGen, despite its intelligent scheduling, struggles to handle the ultra-low inter-region bandwidth inherent in decentralized environments. The communication time between consecutive pipeline stages often dominates GPU computation time, leading to inflated tail latency. By contrast, PARALLAX explicitly incorporates these decentralized communication constraints into its scheduling, enabling globally optimal placement decisions that balance computation with inter-GPU communication overheads. This design allows PARALLAX to consistently deliver lower tail latency, especially under high request rates where communication bottlenecks dominate.

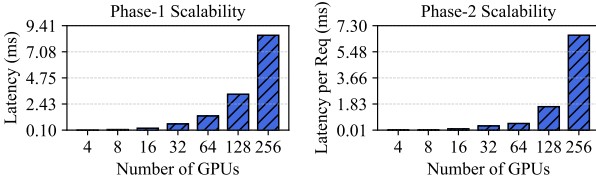

Figure 5: Phase-1 and phase-2 algorithm running time when scaling from smaller clusters (e.g., 4 GPUs) to larger clusters (e.g., 256 GPUs).

**Throughput comparison between PARALLAX and HexGen.** With respect to throughput, PARALLAX consistently maintains higher request processing rates compared to HexGen. For FP16 models, the performance gains are substantial: on WildGPT with a request rate of 32, PARALLAX achieves 0.40 req/s compared to HexGen's 0.11 req/s, representing a 3.6× improvement. Under lighter FP8 models where both systems experience reduced communication constraints, PARALLAX continues to demonstrate superior performance—for instance, on ShareGPT with a request rate of 8, throughput increases from 0.33 req/s to 0.43 req/s. These findings demonstrate that PARALLAX not only reduces tail latency but also maximizes throughput through joint optimization of computation and communication in decentralized environments.

### 4.3 SCHEDULING ALGORITHM ANALYSIS (RQ2)

**Scheduling efficiency and scalability.** Both phases of our scheduler demonstrate lightweight operation and scale effectively to large, decentralized clusters. Phase-1 (model allocation) executes once per configuration and maintains single-digit millisecond latency even at 256 GPUs: execution time increases from 0.10 ms (4 GPUs) to 8.55 ms (256 GPUs). This efficiency is attributed to the region-based and latency-dominant heuristics that substantially reduce the search space. Phase-2 (per-request GPU pipeline chain selection) scales with the number of candidate copies per layer, consistent with the $\mathcal{O}(L\bar{R}^2)$ complexity: execution time increases from 0.0136 ms/req at 4 GPUs to 6.6339 ms/req at 256 GPUs. The absolute computational overhead of this phase remains within low single-digit milliseconds per request, which is negligible relative to end-to-end inference latencies. Importantly, the design decisions from Phase-1 (contiguous layer slices, region-bounded placement) effectively reduce $\bar{R}$ in Phase-2, enabling the dynamic-programming sweep to operate on a compact DAG and converge in a single pass, thereby facilitating real-time chain selection and effective load balancing under heterogeneous, cross-region bandwidth constraints.

**Impact on handling dynamic membership.** Efficient scheduling is essential in decentralized environments with dynamic GPU membership. Phase-1 allocation completes within 10 ms at 256 GPUs ($0.10\,\text{ms} \rightarrow 8.55\,\text{ms}$), enabling cost-effective global reconfiguration, while Phase-2 incurs minimal per-request overhead ($0.0136\,\text{ms/req} \rightarrow 6.6339\,\text{ms/req}$) for real-time path selection. These low overheads allow the system to maintain stable performance despite cluster dynamics while preserving globally efficient allocations.

## 5 CONCLUSION

In this paper, we present PARALLAX, a prototype system, to demonstrate that decentralized collaborative peer-to-peer GPU pools can provide practical LLM inference when equipped with principled resource and request scheduling. Our two-phase scheduling design addresses the NP-hard joint placement-and-routing problem under extreme heterogeneity for LLM inference service, where Phase 1 scheduling determines model allocation via dynamic programming with region-aware, latency-dominant heuristics and water-filling, and Phase 2 scheduling decides request-time GPU pipeline chain selection via a DAG dynamic program over a DHT of live per-layer latencies and inter-GPU RTTs. Our empirical study illustrates that PARALLAX significantly improves end-to-end latency, throughput, and SLO attainment relative to decentralized baselines. We believe the prototype system PARALLAX suggests that decentralized LLM inference service shows great potential: careful placement and online chain selection can convert diverse, low-bandwidth GPU resources into an efficient LLM inference serving platform.

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

| Symbol | Meaning | Type / Unit | Notes |
|---|---|---|---|
| $L$ | Number of model layers | $\mathbb{N}$ (layers) | Layers indexed by $\{1, \ldots, L\}$. |
| $\mathbf{G}$ | Set of GPUs | $\{g_1, \ldots, g_N\}$ | Decentralized, heterogeneous devices. |
| $N$ | Number of GPUs | $\mathbb{N}$ | $N = |\mathbf{G}|$. |
| $\mathcal{R}$ | Set of regions | $\{r_1, \ldots, r_{|\mathcal{R}|}\}$ | Region-based scheduling scope in Phase 1. |
| $\mathbf{G}^{(r)}$ | GPUs in region $r$ | subset of $\mathbf{G}$ | Phase 1 allocation constrained within regions. |
| $g_i$ | The $i$-th GPU | element of $\mathbf{G}$ | Device identifier. |
| $c_i$ | Layer capacity of $g_i$ | $\mathbb{N}$ (layers) | Max transformer layers in VRAM (with margin). |
| $F_i$ | Compute capacity of $g_i$ | $\mathbb{R}^+$ (FLOPs/s) | Used by water-filling for intra-pipeline balance. |
| $\mathbf{c}$ | Sorted capacity vector | $(c_1 \geq \cdots \geq c_N)$ | Non-increasing sort of $\{c_i\}$. |
| $k$ | Number of pipeline replications | $\mathbb{N}$ | Concurrent pipelines serving requests. |
| $k_{\max}$ | Max feasible replications | $\mathbb{N}$ | $k_{\max} = \min\left(N, \lfloor \sum_i c_i/L \rfloor\right)$. |
| $s^\star(k)$ | Min total stages for $k$ replications | $\mathbb{N}$ (stages) | Phase 1 DP objective. |
| $s^\star(k)/k$ | Avg stages per replication | $\mathbb{R}^+$ (stages/replica) | Proxy for per-replication latency. |
| $\mathrm{dp}_1(i, \mathbf{r}, f)$ | Phase 1 DP state | — | At index $i$: residuals $\mathbf{r}$ and completed pipelines $f$. |
| $i$ | Index in $\mathbf{c}$ | $\{1, \ldots, N\}$ | Cursor over sorted GPUs. |
| $\mathbf{r}$ | Residuals of partial pipelines | nondecreasing tuple | $\mathbf{r} = (r_1 \leq \cdots \leq r_m)$, each $r_j \in \{1, \ldots, L-1\}$. |
| $r_j$ | Residual for pipeline $j$ | $\{1, \ldots, L-1\}$ (layers) | Pipeline completes when $r_j \leq 0$. |
| $f$ | Fully assigned pipelines | $\mathbb{N}$ | Count of pipelines with all $L$ layers. |
| $T_{\mathrm{comp}}$ | Avg compute time per replication | time (e.g., ms) | Excludes inter-stage communication. |
| $r_{\mathrm{RTT}}$ | Avg inter-stage hop latency | time (e.g., ms) | Mean one-way RTT between consecutive stages. |
| $\alpha$ | Replication preference exponent | $\mathbb{R}^+$ | Larger $\alpha$ favors throughput via more replications. |
| $Z(k)$ | Phase 1 score at replication $k$ | $\mathbb{R}^+$ | $Z(k) = k^\alpha / \left(T_{\mathrm{comp}} + (s^\star(k)/k)\, r_{\mathrm{RTT}}\right)$. |
| $x_i$ | Fractional layer allocation of $g_i$ | $\mathbb{R}_{\geq 0}$ (layers) | Water-filling: $x_i = \min(c_i, \lambda F_i)$. |
| $\lambda$ | Water-filling scale | $\mathbb{R}_{>0}$ | Found by binary search to make $\sum_i x_i = L$. |
| $x_i^{\mathrm{int}}$ | Integer layer allocation of $g_i$ | $\mathbb{N}$ (layers) | Rounded from $x_i$ under $\sum_i x_i^{\mathrm{int}} = L$, $x_i^{\mathrm{int}} \leq c_i$. |
| $(\ell, g)$ | DAG node | pair | Layer $\ell$ replicated on GPU $g$. |
| $\tau_{g,\ell}$ | Profiled latency of layer $\ell$ on $g$ | time (e.g., ms) | Node weight in Phase 2 DAG (from DHT). |
| $\rho_{gg'}$ | One-way link latency $g \to g'$ | time (e.g., ms) | Edge weight in DAG; measured periodically. |
| $d_{\ell,g}$ | Min cumulative latency to $(\ell, g)$ | time (e.g., ms) | Phase 2 DP cost table entry. |
| $\mathrm{dp}_2(\ell, g)$ | Phase 2 DP state | time (e.g., ms) | Minimum latency to reach layer $\ell$ on GPU $g$. |
| $\hat{g}$ | Argmin terminal GPU at layer $L$ | element of $\mathbf{G}$ | $\hat{g} = \arg\min_g d_{L,g}$ for backtracking. |
| $\bar{R}$ | Avg number of replications per layer | $\mathbb{R}^+$ | Candidate copies per layer admitted into DAG. |
| DHT | Distributed hash table | — | Stores live $\{\tau_{g,\ell}\}$, $\{\rho_{gg'}\}$; refreshed every 1–2 s. |
| RTT | Inter-GPU latency (one-way) | time (e.g., ms) | Used synonymously with $\rho_{gg'}$. |
| $\mathcal{O}(L\bar{R}^2)$ | Phase 2 time complexity | — | Each DAG edge relaxed once in a forward sweep. |
| $\mathcal{O}(L\bar{R})$ | Phase 2 space complexity | — | DP table plus back-pointers. |
| $c_{\mathrm{new}}$ | Capacity of a newly joined GPU | $\mathbb{N}$ (layers) | Inserted into $\mathbf{c}$ on membership change. |

Table 1: Summary of notation for the two-phase decentralized scheduling algorithm.

# A   NOTATION SUMMARY

We present a summary of the notation used in our scheduling algorithm in Table 1.

# B   THE USE OF LLMS IN WRITING

We used LLM, namely OPENAI-GPT5, to polish the writing of this manuscript. No other generative AI functionality is used in the writing of this submission.

