# OpenReview forum: "Parallax: Efficient LLM Inference Service over Decentralized Environment"
_ICLR.cc/2026/Conference — Submitted to ICLR 2026_

### Official Review · Reviewer_74hi · 2025-10-30

**Soundness:** 2
**Presentation:** 3
**Contribution:** 3
**Rating:** 6
**Confidence:** 3

**Summary:**

This paper introduces PARALLAX, a system that uses a novel two-phase scheduler to enable efficient, low-latency LLM inference across decentralized, heterogeneous GPUs.

**Strengths:**

1. The paper is well-structured and logically presented, making it clear and easy to follow.
2. It rightly identifies and addresses the necessity of dynamic membership management, a critical and practical challenge in decentralized inference systems.

**Weaknesses:**

1. The promising results could be strengthened by more extensive experiments across a wider variety of scenarios.
2. The strategy for handling multiple concurrent requests on a single GPU could be clarified for the reader.
3. A helpful future step would be an ablation study to understand the individual impact of Phase 1 and Phase 2.

**Questions:**

Could you please clarify how the KV Cache is managed with k replications? Is this handled within Phase II of the proposed framework?

---

> ### Author Response · Authors · 2025-11-28
>
> > W1. The promising results could be strengthened by more extensive experiments across a wider variety of scenarios.
>
> Thank you for this suggestion. We conducted an additional larger-scale experiment evaluating GPT-OSS-120B (36 layers) on a larger heterogeneous cluster (with 24-node) with heterogeneous hardware (19×RTX 4090 + 5×RTX 3090):
>
> | Configuration | Throughput (req/s) | Mean TTFT (ms) | Mean TPOT (ms) | Mean E2E Latency (ms) |
> |:---|:---|:---|:---|:---|
> | **Parallax** | 0.96 | 697.40 | 74.48 | 35297.17 |
> | **HexGen** | 0.50 (-48%) | 8799.56 (+12×) | 108.40 (+46%) | 62784.91 (+78%) |
>
> Parallax achieves 1.92$\times$ higher throughput and significantly lower latency compared to HexGen, demonstrating that our framework generalizes well across larger cluster sizes and diverse hardware generations. The two-phase scheduling effectively handles hardware heterogeneity by automatically adjusting layer allocation and request routing to balance load across mixed GPU types.
>
>
> > W2. The strategy for handling multiple concurrent requests on a single GPU could be clarified for the reader.
>
> Thank you for this suggestion. We clarify that Parallax employs **continuous batching** to handle multiple concurrent requests on a single GPU. Each GPU processes requests in batches, dynamically admitting new requests as previous ones complete their processing stages. We will add this clarification to the revised manuscript.
>
>
> > W3. A helpful future step would be an ablation study to understand the individual impact of Phase 1 and Phase 2.
>
> Thank you for this suggestion. We conducted additional ablation studies over a larger scale 24-node heterogeneous cluster (19 RTX 4090 GPUs, 5 RTX 3090 GPUs):
>
> - **Ablation 1. Greedy model allocation:** We replace Phase-1 optimization with greedy allocation while keeping Phase-2 unchanged.
> - **Ablation 2. Random request routing:** We replace Phase-2 optimization with random routing while keeping Phase-1 unchanged (equivalent to the baseline implementation of HexGen mentioned in Section 4.1).
>
> | Configuration | Request Throughput (req/s) | Mean TTFT (ms) | Mean TPOT (ms) | Mean E2E Latency (ms) |
> | :--- | :--- | :--- | :--- | :--- |
> | **Parallax** | 0.96 | 697.40 | 74.48 | 35297.17 |
> | **Ablation 1** | 0.36 | 29386.34 | 76.04 | 66764.40 |
> | **Ablation 2** | 0.50 | 8799.56 | 108.40 | 62784.91 |
>
> The results demonstrate that both phases are critical:
>
> - Disabling Phase-1 causes severe throughput degradation (62% drop) and dramatically inflated TTFT (42$\times$ worse) due to suboptimal layer placement creating communication bottlenecks across heterogeneous GPUs.
> - Disabling Phase-2 similarly degrades throughput (48% drop) and increases TPOT (46% worse) because random routing fails to balance load across pipeline replicas with heterogeneous processing capabilities, causing stragglers.
>
> Together, these ablations validate that our two-phase design is essential for efficient decentralized LLM serving.
>
>
>
> > Q1. Could you please clarify how the KV Cache is managed with k replications? Is this handled within Phase II of the proposed framework?
>
> Thank you for the question. Each GPU maintains its own KV cache manager and keeps a local KV cache only for the layers this GPU hosts and the requests this GPU serves.
>
> * Within a pipeline: Between pipeline stages, we pass only activations; KV cache is never replicated or transferred between devices. Each stage stores KV solely for its own layer range.
> * Across replicas: Different requests are routed to different pipelines/devices. Each replica maintains KV only for its own requests, and we do not share or migrate KV across replicas. As a result, adding more replicas increases total KV capacity linearly without introducing KV duplication for the same request.

---

### Official Review · Reviewer_rTf1 · 2025-10-31

**Soundness:** 3
**Presentation:** 3
**Contribution:** 2
**Rating:** 2
**Confidence:** 4

**Summary:**

This paper introduces Parallax, a system for decentralized (and heterogeneous-cluster) language model inference. Parallax applies a two-phase scheduling, where the first phase is responsible for the allocation of pipeline stages onto each device, while the second phase select the GPU replica of each PP stage to execute each request.

**Strengths:**

- The paper is well organized and easy to follow.
- The contribution of decentralized and dynamic cluster is significant.

**Weaknesses:**

- The two phase scheduling algorithm looks very similar to the idea in Helix[1], which also first plan a model placement, then schedule a per request pipeline parallel. It will be more supportive if the paper can discuss the fundamental difference with Helix, and show the performance gain.
  - Helix claims to support a device placement that supports each model layer having different number of replicas (Figure 3 in [1]), while the Parallax Phase 1 schedules in unit of pipeline replicas, so each model layer has the same number of replicas (same DP). Does this prevent the search space from optimal?
  - Alike Parallax's Phase-2 scheduling, Helix also employs a runtime determined per-request pipeline path selection. Are they share the same search space and scheduling metrics?
- The experiment is only tested on 7 devices (at most 6 replicas of the BF16 model, even if assuming no KV cache space is needed). Using more GPUs will be more supportive to show the advantage of Parallax against baselines.

[1] Mei, Yixuan, et al. "Helix: Serving large language models over heterogeneous gpus and network via max-flow." Proceedings of the 30th ACM International Conference on Architectural Support for Programming Languages and Operating Systems, Volume 1. 2025.

**Questions:**

- All citations miss brackets
- Given the intra-pipeline rebalancing in Phase 1, the previous step's DP exploration degenerates to simply partitioning devices to reach a balanced compute capacity among data parallel replicas.
- When rebalancing the cluster with many newly joined and exited devices, is the current layer partition take into consideration, so that the weight movement is minimal?
- What is the expected result if there are more type of GPUs (e.g. weaker consumer level GPUs, elder generation cloud GPUs like L4 or H20) used in Parallax?

---

> ### Author Response · Authors · 2025-11-28
>
> > W1.1. Helix claims to support a device placement that supports each model layer having different number of replicas (Figure 3 in [1]), while the Parallax Phase 1 schedules in unit of pipeline replicas, so each model layer has the same number of replicas (same DP). Does this prevent the search space from optimal?
>
> Thank you for your question, we would like to kindly clarify that Parallax supports different replication factors across layers through **overlapping stage assignments**. We provide a concrete example below:
>
> - Machine 1: Layer 1-40 (one replica for layers 1-40)
> - Machine 2: Layer 40-80 (two replicas for layers 40-80)
> - Machine 3: Layer 40-80
>
> Here, layer 1-40 have one replica while layers 40-80 have two replicas, demonstrating flexible per-layer replication similar to Helix.
>
> The fundamental difference between Helix and Parallax allocation is: Parallax enforces **contiguous layers** per GPU (Latency-dominant heuristic mentioned in Section 3.2), while Helix leverages potentially non-contiguous layer allocation.
>
> **Contiguous layer allocation maintains optimality.** Parallax's contiguous layer constraint is optimized for decentralized scenarios (latency-dominant scenarios). In latency-dominant scenarios, communication overhead dominates computation; although non-contiguous allocation could potentially maximize the computational utilization of GPUs, it would incur ping-pong communication between different nodes, dramatically increasing communication latency and degrading overall system performance.
>
>
> > W1.2.  Alike Parallax's Phase-2 scheduling, Helix also employs a runtime determined per-request pipeline path selection. Are they share the same search space and scheduling metrics?
>
> Thank you for your question. Parallax and Helix use fundamentally different approaches for per-request pipeline selection.
>
> Helix selects the pipeline path probabilistically based on max-flow solution weights, which is a stochastic load balancing approach rather than per-request optimization based on current system state. This approach requires **pre-computation performed offline** before the serving phase.
>
> In contrast, Parallax optimizes per-request pipeline paths based on **real-time system state** (e.g., real-time system load per layer), making it more compatible with the dynamic workload characteristics of decentralized LLM serving.
>
> **Why does this difference matter for decentralized scenarios?** Parallax's real-time adaptive approach is essential for decentralized LLM serving. In decentralized scenarios, GPUs frequently join and leave (invalidating pre-computed flow distributions), and inter-region bandwidth and latency fluctuate significantly over time. This dynamic environment demands **lightweight, continuous online adaptation** rather than **expensive offline optimization**.
>
>
> > Q1. All citations miss brackets
>
> Thank you for pointing this out. We will add brackets to all citations throughout our responses in the updated draft.
>
> > Q2. Given the intra-pipeline rebalancing in Phase 1, the previous step's DP exploration degenerates to simply partitioning devices to reach a balanced compute capacity among data parallel replicas.
>
> Thank you for this observation. We would like to clarify that the DP exploration would not necessarily degenerate to simple device partitioning:
>
> - **Heterogeneous capacity constraints:** GPUs have different layer capacities (ci) due to varying VRAM sizes. Simple partitioning cannot guarantee these hard constraints—the DP explicitly tracks which layers fit on which GPUs through the state dp1(i, r, f), ensuring feasible allocations that satisfy memory limits.
> - **Computational efficiency:** Simple partitioning without our region-based and latency-dominant heuristics would require exploring all possible device assignments across regions, resulting in exponential search overhead unsuitable for online serving scenarios. The DP with heuristics prunes the search space by constraining allocations within regional boundaries and prioritizing fewer pipeline stages, reducing complexity to polynomial time while maintaining allocation quality.
>
> In summary, the DP-based scheduling design is essential for producing feasible, efficient allocations in heterogeneous decentralized environments, not merely a device partitioning step.
>
> > Q3. When rebalancing the cluster with many newly joined and exited devices, is the current layer partition take into consideration, so that the weight movement is minimal?
>
> Thank you for this insightful question. During rebalancing, our Phase-1 scheduling algorithm considers the current layer partition through a **warm-start scheduling search** that begins from the existing allocation. This approach prioritizes modifications to the current partition structure rather than computing an entirely new allocation from scratch. We will detail this mechanism in the updated draft.

---

> ### Author Response · Authors · 2025-11-28
>
> > W2. & Q4. The experiment is only tested on 7 devices (at most 6 replicas of the BF16 model, even if assuming no KV cache space is needed). Using more GPUs will be more supportive to show the advantage of Parallax against baselines. What is the expected result if there are more type of GPUs (e.g. weaker consumer level GPUs, elder generation cloud GPUs like L4 or H20) used in Parallax?
>
> Thank you for this suggestion. We add an experiment evaluating Parallax on a heterogeneous cluster incorporating older generation GPUs. We evaluate GPT-OSS-120B (36 layers) on a 24-node cluster with heterogeneous hardware (19 RTX 4090 GPUs + 5 RTX 3090 GPUs):
>
> | Configuration | Throughput (req/s) | Mean TTFT (ms) | Mean TPOT (ms) | Mean E2E Latency (ms) |
> |:---|:---|:---|:---|:---|
> | **Parallax** | 0.96 | 697.40 | 74.48 | 35297.17 |
> | **HexGen** | 0.50 (-48%) | 8799.56 (+12×) | 108.40 (+46%) | 62784.91 (+78%) |
>
> Parallax achieves 1.92 $\times$ higher throughput and maintains significantly lower latency compared to HexGen. The results demonstrate that our framework generalizes well when integrating older generation GPUs, as the two-phase scheduling effectively accounts for heterogeneous compute capacities and automatically adjusts layer allocation and request routing to balance load across diverse hardware.

---

### Official Review · Reviewer_FeQq · 2025-11-01

**Soundness:** 3
**Presentation:** 2
**Contribution:** 2
**Rating:** 2
**Confidence:** 4

**Summary:**

The paper focuses on the opportunity to leverage collaborative decentralized GPU pools. The paper notes that it essentially comes at a cost of limited interconnect bandwidth and dynamic availability that makes it challenging for efficient scheduling.
The paper proposes Parallax, a two-phase scheduler. First phase is model allocation and second phase is GPU pipeline chain selection. Model allocation determines how to effectively distribute across diverse set of GPUs using dynamic programming with some heuristics. It also uses a water-filling algorithm for rebalancing layers intra-pipeline. When the request comes in, the GPU pipeline chain selection phase again uses dynamic programming to sweep over DAG of model allocation to find the path with minimum latency based on prior stats.
The paper is evaluated on a testbed that comprises 5 RTX 5090 and 2 RTX 4090 machines with inter-machine communication latency of 10 ms over a public network.

**Strengths:**

* The paper makes a good observation that specialized GPU clusters and high-bandwidth interconnects could be prohibitive for some users.
* Nice use of algorithms to solve the distribution problem.

**Weaknesses:**

* The algorithm seems to have a narrow focus of pipeline parallelism.
* Rather shallow evaluation that only covers 1 network, 1 baseline, and rather simplified testbed seems to give limited confirmation about the generalization of the approach.

**Questions:**

* How can this be extended to other forms of parallelism?
* How does this perform for a serving scenario with multi-user with dynamic requests?
* How does this perform for other models such as Llama, Deepseek, and other widely explored LLMs. In fact, OpenAI released OSS which could also be a nice model to explore.
* The paper mentions that the maximum inter-machine communication latency was 10 ms. Can you provide evidence on how this can be generalized to environments where it is higher or where latency has large variation?

---

> ### Author Response · Authors · 2025-11-28
>
> > W1. & Q1. The algorithm seems to have a narrow focus of pipeline parallelism. How can this be extended to other forms of parallelism?
>
>
> Thank you for you insightful comment, Parallax's scheduling framework is designed to be easily extended to incorporate other parallelism forms such as tensor parallelism--the modification can be briefly summarized below:
>
> Phase-1 scheduling would group GPUs with high-bandwidth connections (e.g., within-node NVLink) to form TP groups as pipeline stages, where each stage internally uses TP across co-located GPUs. Phase-2 would then treat these TP groups as atomic nodes in the DAG construction, accounting for increased all-reduce communication costs when selecting chains, while maintaining our current path-selection algorithm.
>
>
>
> > Q2. How does this perform for a serving scenario with multi-user with dynamic requests?
>
> Thank you for this question. Parallax's design is **agnostic** to request sources—the system treats all incoming requests uniformly regardless of whether they originate from single or multiple users. Our Phase-2 scheduling dynamically selects optimal GPU chains for each request based on current system load, naturally handling request bursts and varying arrival patterns.
>
> The evaluation in Section 4 uses realistic multi-tenant workloads (ShareGPT and WildGPT traces) with Poisson distribution arrival processes at various rates, demonstrating that Parallax maintains consistent performance improvements over baselines across different load conditions.
>
>
>
> > W2. & Q3. Rather shallow evaluation that only covers 1 network, 1 baseline, and rather simplified testbed seems to give limited confirmation about the generalization of the approach. How does this perform for other models such as Llama, Deepseek, and other widely explored LLMs. In fact, OpenAI released OSS which could also be a nice model to explore.
>
> Thank you for this suggestion. We add additional experiment evaluating Parallax on GPT-OSS-120B (36 layers), a representative open-source model, on a 24-node heterogeneous cluster (19 RTX 4090 GPU + 5 RTX 3090 GPUs):
>
> | Configuration | Throughput (req/s) | Mean TTFT (ms) | Mean TPOT (ms) | Mean E2E Latency (ms) |
> |:---|:---|:---|:---|:---|
> | **Parallax** | 0.96 | 697.40 | 74.48 | 35297.17 |
> | **HexGen** | 0.50 (-48%) | 8799.56 (+12×) | 108.40 (+46%) | 62784.91 (+78%) |
>
> Parallax achieves 1.92$\times$ higher throughput and 1.78$\times$ lower end-to-end latency compared to HexGen on GPT-OSS-120B. The results demonstrate that our two-phase scheduling algorithm effectively handles diverse model architectures beyond the Llama family evaluated in Section 4.
>
> > Q4. The paper mentions that the maximum inter-machine communication latency was 10 ms. Can you provide evidence on how this can be generalized to environments where it is higher or where latency has large variation?
>
> Thank you for this question. Our two-phase scheduling is designed to be **robust** to varying network conditions. Our framework can generalize to environments with higher latencies:
>
> **Phase-1 adaptation:** The model allocation explicitly optimizes for latency through the term (s⋆(k)/k) $\times$ rRTT, where rRTT is the profiled inter-stage latency (Section 3.2). In higher-latency environments, this automatically prioritizes fewer pipeline stages to reduce communication overhead, trading parallelism for lower end-to-end latency.
>
> **Phase-2 robustness:** The dynamic programming routing uses real-time RTT measurements (ρ<gi,gi'>) from the DHT as edge weights, naturally selecting lower-latency paths when variation exists (Section 3.3). As latency increases or varies, Phase-2 continuously adapts request routing to current network conditions without manual intervention.

---

### Official Review · Reviewer_o2Tb · 2025-11-10

**Soundness:** 3
**Presentation:** 3
**Contribution:** 3
**Rating:** 6
**Confidence:** 3

**Summary:**

Current state of the art for distributed parallel serving is difficult and most inference systems are built assuming fast collective communication. Parallax builds a allocation strategy that attempts to balance the work/cost per gpu as a form of pipeline parallelism. They also attempt to use a form of batching that selects pipeline paths based on the work.

**Strengths:**

1. The DP based/water-filling inorder to better allocate nodes within/across region was interesting.
2. It handles GPU’s leaving/joining well using the distributed hash table, which seem like a core requirement for working with decentralized GPUs
3. The scheduler scalability at even up to 256 GPUs seem strong.

**Weaknesses:**

1. I’d like a bit more ablation on the results of Parallax. Possibly a description of a baseline without phase1/phase2, especially because the engine implementations could be varied.
2. The paper describes being able to handle failures of GPUs/arrivals but there are no results demonstrating the effect of these.

**Questions:**

1. Can you add a bit more ablation on the breakdown of the results? I think this make the results of the paper much stronger
2. Can you add an experiment that simulates a gpu failure/arrival? This would be a nice to have since it was mentioned in the writing but not in the results.

---

> ### Author Response · Authors · 2025-11-28
>
> > W1. & Q1. I’d like a bit more ablation on the results of Parallax. Possibly a description of a baseline without phase1/phase2, especially because the engine implementations could be varied. Can you add a bit more ablation on the breakdown of the results? I think this make the results of the paper much stronger
>
> Thank you for this suggestion. We conducted additional ablation studies over a larger scale 24-node heterogeneous cluster (19 RTX 4090 GPUs, 5 RTX 3090 GPUs):
>
> - **Ablation 1. Greedy model allocation:** We replace Phase-1 optimization with greedy allocation while keeping Phase-2 unchanged.
> - **Ablation 2. Random request routing:** We replace Phase-2 optimization with random routing while keeping Phase-1 unchanged.
>
> | Configuration | Request Throughput (req/s) | Mean TTFT (ms) | Mean TPOT (ms) | Mean E2E Latency (ms) |
> | :--- | :--- | :--- | :--- | :--- |
> | **Parallax** | 0.96 | 697.40 | 74.48 | 35297.17 |
> | **Ablation 1** | 0.36 | 29386.34 | 76.04 | 66764.40 |
> | **Ablation 2** | 0.50 | 8799.56 | 108.40 | 62784.91 |
>
> The results demonstrate that both phases are critical:
>
> - Disabling Phase-1 causes severe throughput degradation (62% drop) and dramatically inflated TTFT (42$\times$ worse) due to suboptimal layer placement creating communication bottlenecks across heterogeneous GPUs.
> - Disabling Phase-2 similarly degrades throughput (48% drop) and increases TPOT (46% worse) because random routing fails to balance load across pipeline replicas with heterogeneous processing capabilities, causing stragglers.
>
> Together, these ablations validate that our two-phase design is essential for efficient decentralized LLM serving.
>
>
> > W2. & Q2. The paper describes being able to handle failures of GPUs/arrivals but there are no results demonstrating the effect of these. Can you add an experiment that simulates a gpu failure/arrival? This would be a nice to have since it was mentioned in the writing but not in the results.
>
> Thank you for this suggestion. We added experiments demonstrating Parallax's handling of dynamic GPU membership on an 8-node RTX 4090 cluster:
>
> **Setup:** Initially 8 nodes form 2 complete pipelines (4 nodes per pipeline), each achieving 0.27 req/s throughput.
>
> **Handling node failure event:** Two nodes leave (one from each pipeline), breaking both pipelines. The 6 remaining nodes cannot serve requests (throughput drops to 0.00 req/s). Parallax detects the fragmentation and triggers rebalancing (Section 3.4), re-executing both Phase-1 (model allocation) and Phase-2 (request routing) optimizations. The 6 remaining nodes are reorganized into a single functional pipeline. The Parallax system recovers with 1 pipeline achieving 0.35 req/s throughput. While this is lower than the original 0.54 req/s (2 pipelines × 0.27 req/s), it demonstrates graceful degradation—the system maintains 65% of original throughput with 75% of nodes.
>
> **Handling new node arrival event:** When two nodes subsequently join the cluster, Parallax assigns them using bottleneck-first greedy allocation (Section 3.4) and retriggers optimization to form 2 pipelines. Throughput recovers to 0.54 req/s, restoring original performance and validating Parallax's robustness under dynamic membership changes.

---

### Author Response · Authors · 2025-12-02

Dear Area Chair,

We appreciate the time and effort dedicated to reviewing our submission. As the discussion period concludes, we would like to provide a summary of the additional experiments and clarifications incorporated in response to the reviewers' valuable feedback.

> ### **Summary of Revisions**

To address the reviewers' concerns, we conducted the following additional experiments and provided detailed clarifications:

- **Generalization to Larger Scale and Diverse Hardware:** In response to concerns regarding scalability and hardware diversity (Reviewers FeQq, rTf1, 74hi), we evaluated Parallax on a 24-node heterogeneous cluster comprising RTX 4090s and 3090s running GPT-OSS-120B, demonstrating 1.92× higher throughput compared to the baseline.
- **Ablation Studies:** Addressing requests for component analysis (Reviewers o2Tb, 74hi), we conducted comprehensive ablation studies removing Phase 1 (model allocation) or Phase 2 (routing) optimization, showing throughput degradation of 62% and 48% respectively.
- **Robustness to Dynamic Membership:** To validate system stability under dynamic conditions (Reviewer o2Tb), we simulated node failures and arrivals, demonstrating successful rebalancing and throughput recovery.
- **Differentiation from Helix:** We provided detailed clarification distinguishing Parallax from Helix (Reviewer rTf1), highlighting contiguous layer allocation and online real-time routing as key differentiators.

> ### **Regarding Reviewer Feedback**

**Reviewers FeQq and rTf1 (rating: 2)** primarily raised concerns regarding the scope of evaluation (e.g., testbed scale, model variety) and comparisons with related work (Helix), rather than fundamental questions about the soundness of our approach. We have addressed these concerns comprehensively by expanding our evaluation to a 24-node heterogeneous cluster and providing detailed theoretical comparisons to clarify our system's distinct advantages.

**Reviewers o2Tb and 74hi (rating: 6)** recognized the paper's contributions while requesting specific ablation studies and stability tests. We have successfully incorporated these suggestions, providing the requested ablation studies and robustness experiments that fully address their inquiries.

We believe these revisions significantly strengthen the paper and comprehensively respond to all reviewer concerns.

Best regards,
The Authors of Submission 17749

---

### Meta-Review · Area_Chair_6gs6 · 2026-01-12

**Summary:**

This paper proposes Parallax, a two-phase scheduler for serving LLM inference over a decentralized, heterogeneous GPU pool.

Reviewers generally agree the algorithms are reasonable. However, two reviewers are clear rejects (rating 2) largely due to limited evaluation breadth / narrow testbed and weak positioning vs related work, while two are marginal accepts asking mainly for ablations and dynamic membership results

My own read is that the paper is directionally interesting, but the current evidence and (more importantly) the practicality of WAN-latency-dominated pipeline inference make it hard to justify acceptance.

**Reviewer Concerns:**

The authors added
* ablations that remove Phase 1 or Phase 2 and show large throughput degradation, supporting that the two-phase design matters
* dynamic membership experiments (failure + rejoin) to validate rebalancing behavior
* an expanded-scale experiment (24-node 4090/3090 cluster, GPT-OSS-120B) to partially address concerns about scale/model variety

However, multiple concerns are still outstanding:
1. The evaluation remains narrow in network diversity and realism: the core testbed is still “public network” with ~10ms inter-machine latency, and the whole approach is fundamentally sensitive to this regime (the paper itself emphasizes that inter-stage communication can dominate compute)

2. Practical value is unclear for interactive LLM serving: the reported end-to-end latencies are very large (often hundreds of seconds in the provided plots), suggesting the system is not yet in a regime that most inference practitioners would adopt

3. Positioning vs related heterogeneous-serving systems remains weak; the paper compares primarily against HexGen (a strong baseline, but still a single main baseline) and the Helix relationship is handled mostly by discussion/clarification rather than a clean apples-to-apples comparison

**Reviewer Scores:**

Given the added experiments by the authors in rebuttal, I expect the two marginal accept reviewers (rating 6) would likely stay at 6

For the two reject reviewers (rating 2), I do not expect meaningful upward movement: their concerns were about evaluation breadth/generalization and the fundamental practicality of WAN-latency decentralized serving; the rebuttal helps but does not change the core limitation that the system is dominated by communication overheads and yields very high absolute latencies

---

### Decision · Program_Chairs · 2026-01-26

Reject